# Machine-Learning-Based Scoring System for Antifraud CISIRTs in Banking Environment

Michal Srokosz [1,*], Andrzej Bobyk [2], Bogdan Ksiezopolski [1] and Michal Wydra [3]

1   Polish-Japanese Academy of Information Technology, 02-008 Warsaw, Poland
2   Faculty of Mathematics, Physics and Computer Science, Maria Curie-Skłodowska University in Lublin,
    20-033 Lublin, Poland
3   Department of Computer Science, Lublin University of Technology, 20-618 Lublin, Poland
*   Correspondence: msrokosz@pjwstk.edu.pl

**Abstract:** The number of fraud occurrences in electronic banking is rising each year. Experts in the field of cybercrime are continuously monitoring and verifying network infrastructure and transaction systems. Dedicated threat response teams (CSIRTs) are used by organizations to ensure security and stop cyber attacks. Financial institutions are well aware of this and have increased funding for CSIRTs and antifraud software. If the company has a rule-based antifraud system, the CSIRT can examine fraud cases and create rules to counter the threat. If not, they can attempt to analyze Internet traffic down to the packet level and look for anomalies before adding network rules to proxy or firewall servers to mitigate the threat. However, this does not always solve the issues, because transactions occasionally receive a "gray" rating. Nevertheless, the bank is unable to approve every gray transaction because the number of call center employees is insufficient to make this possible. In this study, we designed a machine-learning-based rating system that provides early warnings against financial fraud. We present the system architecture together with the new ML-based scoring extension, which examines customer logins from the banking transaction system. The suggested method enhances the organization's rule-based fraud prevention system. Because they occur immediately after the client identification and authorization process, the system can quickly identify gray operations. The suggested method reduces the amount of successful fraud and improves call center queue administration.

**Keywords:** intrusion detection system; bank fraud detection; machine learning; CSIRT; IDS architecture; autoencoder

## 1. Introduction

Fraud occurrences and theft and fraud rates are increasing each year in electronic banking. Network infrastructure and transaction systems are constantly monitored and verified by specialists in the field of cybercrime. Organizations employ dedicated threat response teams (CSIRTs), which ensure security and prevent hacker attacks. Banking systems undergo periodic security audits and penetration tests, and increasingly newer security policies are implemented inside the institution to respond to current threats.

Banks are not the main target of cybercriminals, but they are an unaware user without IT knowledge and awareness of the basic principles of security on the Internet. Despite continuous education in the field of the safe use of electronic channels, the number of successful attacks is increasing. In addition to the classic stealing of logins and passwords (phishing), Trojans on mobile phones can intercept OTP SMS codes [1]. Advanced Trojans (such as Zeus) can quickly replace an account entered by a customer with a thief's account in the transaction system.

Recently, organized crime groups have hired call centers with consultants to place random calls to people claiming to be employees of the bank's security department, asking

them to make transfers. The problem is so serious that the banks almost immediately reacted and published press releases in applications for clients about the existing threat and how to avoid it (banks do not use "transfer" methods, so this should worry the user). The increasingly sophisticated and insidious theft attacks mean that the average Internet and electronic banking user is unable to independently ensure the security of his/her personal data and financial resources accumulated in the accounts.

Financial institutions are fully aware of this and invest capital in antifraud software and increase employment in CSIRTs. They are constantly looking for new solutions to be implemented by the organization that will help minimize the risk of losing their clients' funds as a result of theft [2,3]. Specialists from the CSIRT react to new threats to the personal data and money of the clients of their organizations on an ongoing basis. Various methods of threat detection are used. For example, a new type of Trojan appears that was written for a specific bank. The CSIRT investigates the fraud cases, and if the organization has a rule-based antifraud system, the team writes rules to block the threat. Otherwise, they can try to analyze Internet traffic even at the packet level to find anomalies and then add network rules on proxy or firewall servers that neutralize the threat.

The above detection methods work on a post-mortem debugging basis. Therefore, the actual fraud has already occurred, and the banks want to limit the scale of the attack so that other customers are not affected by this type of attack. In the case of 0-day fraud, CSIRTs need time to prepare their defense against the threat, which is a process that runs continuously. From a business point of view, such situations negatively affect the perception of the bank. The media describing cases of theft often describe it as "breaking into a bank", which substantially reduces its credibility and trustworthiness, even if the customer is clearly at fault.

Unfortunately, creating rules is not a solution to all problems. Some transactions fall into "gray" scoring; the bank cannot authorize every gray transaction, because it is impossible due to an insufficient number of employees in the call center. The bank's business estimates the risk and bears its consequences in the form of financial losses.

In this study, we designed a machine-learning (ML)-based scoring system, which introduces early warning against fraud in the banking environment. The proposed system analyzes client logins from the banking transaction system and complements the organization's rule-based antifraud system. The system can find gray operations in the early stage just after the client authentication and authorization process. The proposed method decreases the number of successful fraud occurrences and improves call center queue management.

The architecture and workflow that we designed are innovative compared with existing antifraud systems for several reasons. The main advantage of our method is the flexibility of the architecture, which enables the solution to be quickly implemented in any organization. This is because costly and time-consuming modifications do not need to be introduced to the system's functioning in the organization. The solutions currently used in the banking environment have a specific threat detection workflow, which limits their implementation possibilities. Most often, these are systems that are not prepared for trouble-free connection to the working infrastructure. Our market analysis of the available threat detection systems motivated us to design the most-universal architecture possible. We found that the inability to perform efficient integration and implementation is the most-common reason hindering large organizations from extending their security.

This study provides the following contributions:

1. We developed a new flexible architecture for a scoring system with a machine learning scoring extension in a banking environment.
2. We designed a machine learning model based on data from the early stage in banking transaction processes.
3. We developed two autoencoder (AE) models (shallow and deep), which classify the transactions into white, gray, and black.

## 2. Related Work

In the literature, antifraud systems in the banking environment are described. Here, we briefly characterize the individual approaches of banking antifraud systems, and finally, the major limitations of the current state-of-the art will be indicated.

In [4], the proprietary FraudFind framework was discussed along with its architecture based on the publisher–subscriber architecture. In this case, the publisher (agent) delivers data to the queue, which are then analyzed, processed, and presented in a web interface. The authors committed to implementing the project in the coming years.

In [5], the TinAnt architecture was presented, which, in accordance with the assumptions of real-time performance (considering delays of milliseconds), qualifies transactions as fraudulent. Which modules in the architecture should work "online" and which can be performed "offline" to meet the speed requirements were indicated. The designed architecture is thoughtful and precise. It precisely indicates the positions of the implemented antifraud components and the method of data penetration into the system from the outside. This study is valuable because it shows the actual implementation. This system operates in one of the largest fintech companies: Ant Financial.

In [6], an example of a rule-based expert system (RBESS) was described as a simple implementation of artificial intelligence, which was converted into antifraud rules based on expert knowledge. The authors presented two simple rules and hypothesized that it is better to develop systems based only on artificial intelligence (supervised and unsupervised learning) as they are more promising for future use and devoid of any of the weaknesses of rule-based systems. They presented the proof-of-concept ALIDA system. They showed its architecture and the use of public cloud solutions such as integration with Amazon Web Services. In our opinion, the authors did not present a working system, provided no methods of real integration with banking systems, and showed no awareness that financial institutions in Europe cannot easily use public clouds (legal issues).

No examples exist of a generic architecture in the literature that would allow the analysis of the issue from the general perspective of benefits for banks and financial institutions. The reported studies are based on proof-of-concept issues. They do not elaborate on the issue of fraud, but only signal their occurrence or describe a specific event. No mention has been made of the proper procedure for dealing with cases of suspicious transactions classified as gray due to the small amounts of operations. Currently, scientists specializing in fraud research, rule-based methods, expert systems, or even deep learning systems are focusing their attention on card fraud cases. This means that pretrade aspects, which would reduce the number of thefts if detected early enough, are ignored. For this reason, as an extension of the current research, we designed an early detection method. The proposed early detection method detects suspicious transactions before any real operation is attempted. The studies to date have shown a clear polarization: machine and deep learning must be used to replace rule-based and expert antifraud systems. We think that they should be extended with additional nondeterministic modules based on machine and deep learning.

However, studies on the detection of anomalies among the login attempts within the electronic channel remain scarce, except for the recently studied anomaly detection method using graphs [7]. More recent reviews were performed of (rather general) anomaly detection techniques and methods in the context of both shallow and deep machine learning approaches [8,9]. The authors of the former only briefly mentioned autoencoders (however, the vast number of studies they quoted do not refer at all to the specific issue of detecting illegitimate login attempts), whereas the latter investigated and presented thorough analyses of the most-popular and -effective anomaly detection techniques applied to detect financial fraud, with a focus on highlighting the recent advances in the areas of semisupervised and unsupervised learning.

For example, the researchers in [10] introduced a method that models a network login structure by automatically extracting a collection of login patterns using a variation of the market basket analysis algorithm. They employed an anomaly detection approach to detect

malicious logins that are inconsistent with the enterprise network login structure. In a real scenario simulated attack, their system was able to detect 82% of the malicious logins, with a 0.3% false positive rate. They used a real dataset of millions of logins collected over five months from a global financial company for the evaluation of their work. However, they only analyzed a subclass of malicious logins within enterprise networks, namely those that do not obey certain patterns that are generated by the well-behaving enterprise employees. They were only able to detect malicious logins that were inconsistent with the login structure of the enterprise network, using attributes such as user/source/destination type, user business unit, and the source/destination of the computer's application and location. Although the ideas presented in this paper are interesting in general, the study was limited to one company and one type of enterprise, and therefore, the method cannot be directly applied to the problem of detecting unauthorized logins in the general context.

The problem of distinguishing anomalous from regular events for unlabeled datasets is a challenging task, which is even harder when considering that such a dataset is usually highly imbalanced [11,12]. Several methods exist to solve this problem, involving, for example, various clustering methods, self-organizing maps (SOMs), adaptive resonance theory (ART), artificial neural networks, one-class support vector machine (OCSVM), and many others [13–19]. In general, machine learning techniques (for a survey, see, for instance, [17,20,21]) promise to be at least complementary (or even superior) to traditional rule-based and human-defined methods [22–24]. The latter ones, because of attacks having no constant patterns and a rapidly changing behavior over time, can make them cumbersome, quickly obsolete, and therefore, unsustainable [24].

Autoencoders are a class of ML models that have been previously used for anomaly detection in the financial domain [25], but were solely limited to fraudulent activities, mainly connected with credit card misuse [11,26,27] (see Table 11 in [9] or the reviews provided by [28,29], where the authors briefly describe the advantages and disadvantages of credit card fraud detection methods). The problem of account holder authentication, as an important component of a fraud detection strategy for Internet banking security, was previously mentioned [30], where the authors concluded that, despite the many techniques available for intrusion detection, machine learning (especially neural networks) should further be used to expand upon the models developed for transaction anomaly detection. They also discussed data attributes that should be analyzed via neural networks for intrusion detection, hinting at IP address analysis, user geolocation, and session activation based on the time of day.

## 3. New Antifraud CISIRT Scoring System: Architecture

Figure 1 presents the new architecture of the system supporting CSIRTs in a financial institution (bank). Our approach extends the traditional one with an ML module as the external scoring system, which expands the possibilities of the rule-based expert system. Our solution can act as a proxy before such a system and behind the load balancer. As such, the existing bank architecture does not need to be modified. In the next section, we describe the modules and the data flow in the proposed architecture.

### 3.1. Modules

The architecture presented in Figure 1 includes 12 modules with the new ML-based scoring extension.

Electronic channels: the channel customer service for the bank. Using various electronic channels, the client or the bank's service performs financial and non-financial operations on behalf of the client.

Load balancer: an application that breaks down traffic into individual WAF instances to ensure the high availability of services. The bank's infrastructure and software should handle occasional, sudden, and drastic increases in traffic.

Web application firewall (WAF): responsible for checking if requests from the outside world are technically correct; whether they contain potential attacks; whether the requested

endpoints can be accessed from the given channel; whether the headers are properly signed (HMAC or other algorithm); whether the SSL is correct.

Antifraud subsystem: A classic antifraud system should include a main processing module, a rules editor, rules testing, a validation module, and an interface for administrators and the CSIRT. Expanding the system with an integration subsystem enables the use of additional, external scoring system. General processing is a module responsible for rule-based processing of requests, as a result of which the following business decisions are obtained:

- "OK" (white): The request is forwarded to the core financial system.
- "NOT OK" (black): The system automatically rejects the request and performs the programmed actions (e.g., inform the client about an attempted instruction via the notification subsystem and forward the report to the CSIRT).
- "MANUAL" (grey): The system provides information to the CSIRT about the need to manually verify a given operation.

ML scoring extension (new module): The additional antifraud module proposed here has the main task of supporting the CSIRT through the use of classic statistical methods, as well as ML and deep learning methods. It enables the profiling of customer behavior and the detection of unusual actions for the customer. Potentially gray transactions are sent for manual confirmation, even if they do not exceed the minimum amount specified. This module is presented in detail in Section 4.

CSIRT: the team responsible for protecting users against fraud from inside and outside the organization, which controls the operations performed and reacts to unusual suspicious requests by adding new rules to improve security. The team has access to sensitive data: personal data, a list of enabled authorization tools, installed mobile applications of the bank and devices, and the history of transfers and messages directly received from the core financial service.

Notification subsystem: a separate system supporting the operation of the communication flow in the organization that sends banking messages via text messages (most often integrated with several GSM operators using the SMSC protocol), e-mails, and pushes for mobile devices (integration with Google Firebase and the Apple Push Notification Service). Notifications can be of any sort: 2FA, authorization, or advertising.

Core financial service: the central banking system in the organization responsible for handling general and analytical ledgers, sources of money, current and savings accounts, intrabank settlements, and loans and that conducts full reporting. Optionally, it can be extended with a card system, i.e., debit and credit cards.

Call center: a system that is a work tool for operators who are responsible for telephone contact with customers. In addition to the tasks related to the authorization of operations, they also perform sales and marketing work.

Customer file system: a system containing customer files, which include personal data, information on contracts, customer address and correspondence data, information on signed consents (PSD2 and marketing), and history of contact with the bank.

Auth module: This bank authorization system includes password hashes, the list of client authorization tools, full journal logs/authorization, and the history of authorization tool changes.

Support financial system: additional system supporting the core financial service. It is responsible for verifying the operation limits. Currently, due to the need to support modern payment methods, it is used as a generator of virtual card numbers (e.g., Apple Pay or Google Pay). The core financial service, without the participation of the support financial system, is not able to correctly decode a transaction from the clearing file that was performed with a virtual card.

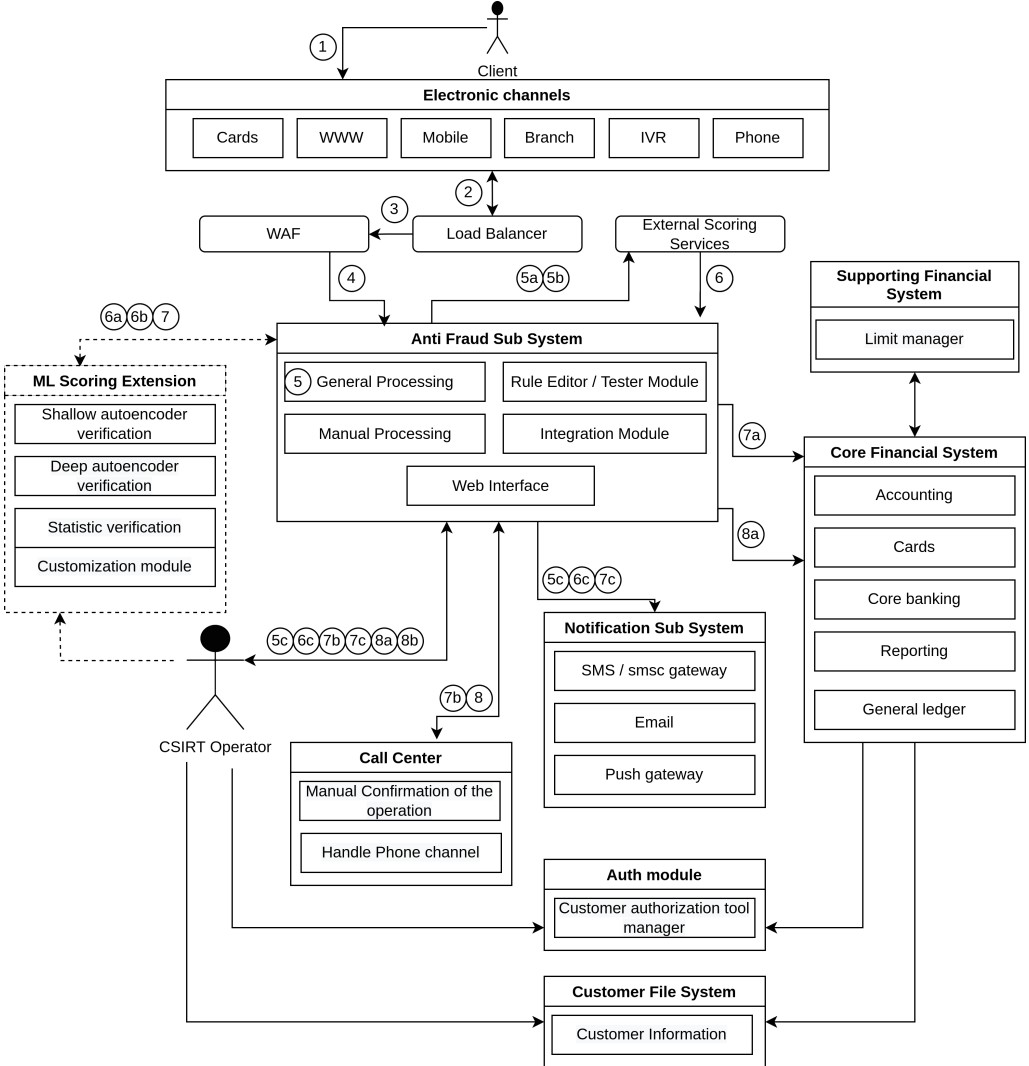

**Figure 1.** Proposed architecture with ML-based scoring module.

### 3.2. Data Flow in Proposed Architecture

The data flow in the proposed architecture is presented in Figure 1, which shows the step numbers that refer to the order of data processing. In this section, these steps are described:

Step 1: The customer or bank service on behalf of the ordering customer through electronic channels performs financial and non-financial operations in the context of the customer.

Step 2: Load balancer receives the request from electronic channels.

Step 3: The received request, owing to the built-in algorithm, is transferred to a selected WAF instance.

Step 4: The WAF verifies the correctness of the received request. In the case of correct verification, the request is transferred to the antifraud system.

Step 5: General processing checks compliance with the rules and makes the following business decisions:

- 5a "OK" (white): The request is sent for extended verification in external scoring services;
- 5b "MANUAL" (grey): The request is sent for extended verification at external scoring services;
- 5c "NOT OK" (black): The request is rejected. The rejection information is sent to the notification subsystem and to the CSIRT.

Step 6: The external scoring service checks compliance with external scoring systems and issues business decisions that are returned to the antifraud subsystem:

- 6a "OK" (white): The request is sent for extended verification to the ML scoring extension (new module);
- 6b "MANUAL" (grey): The request is sent for extended verification to the ML scoring extension (new module);
- 6c "NOT OK" (manual): The request is rejected. The rejection information is sent to the notification subsystem and the CSIRT.

Step 7: The ML scoring extension performs multi-faceted statistical and machine/deep learning verification and issues business decisions that are returned to the antifraud subsystem:

- 7a "OK": The request is valid and passed to the core financial system;
- 7b "MANUAL": The request is directed to the call center for manual verification by the customer, and the CSIRT receives information about the incident;
- 7c "NOT OK": The request is rejected. The rejection information is sent to the notification subsystem and the CSIRT.

Step 8: The call center verifies the client and issues business decisions, which are returned to the antifraud subsystem:

- 8a "OK": The request is valid and passed to the core financial system;
- 8b "NOT OK": The request is rejected and the rejection information is sent to the CSIRT.

*3.3. Data Collection Process: Privacy Leakage*

The storage of sensitive datasets in antifraud systems is an important task. Effective protection is ensured by unique obfuscation and reversibility. The operation scheme is based on generating a unique and real tax ID and a unique and false tax ID (which can be changed back to a real one). The synchronization and cooperation of all systems in the organization means that CSIRTs are able to use both real and false tax IDs. Most often, false tax IDs are used to limit the number of people who have access to real data. This is an additional protection layer of the stored sensitive data. The leakage of confidential data is a high risk, which may lead not only to attempts to steal funds from a specific institution, but also to the theft of identity data to falsify documents for later use in other organizations. The issue of secure data processing is crucial in the area of antifraud systems; it requires the use of the highest standards of current knowledge and technology. Additionally, rigorous compliance with the procedures in the organization can effectively protect sensitive data against illegal use. The possible theft of uniquely obfuscated data is not a threat, because data encrypted in this way are useless.

## 4. ML Scoring Extension

In Figure 2, the ML scoring extension module is presented, which we discuss in detail below.

*4.1. ML Scoring Extension Modules*

In this section, we describe the modules included in the ML scoring module. The architecture of this module with numbers that indicate the steps in the data flow is presented in Figure 2.

External gateways: These entry/access gates to the main ML scoring extension system receive requests and forward them to the integration module in a standardized message. An important aspect is the state of today's IT and technological structure, which obliges new systems to adapt to those already operating in the organization. New systems with additional functionality are implemented without disturbing the current structure. This is because of the value of old (often considered obsolete) systems that are stable, efficient, and free from critical bugs. Their development and modernization would be costly, risky, and

time-consuming. The external gateways module is of key importance for us in future work, because we will analyze login operations and financial operations that we will directly connect with central systems. This will enable us to parallelize and accelerate the issuing of decisions for the organization's main antifraud system.

Integration module: In mature organizations, the supplier is responsible for the implementation of new software. It is their duty to integrate their system with the existing one. The exceptions are fintech/startup companies that most often order a SAAS service, in which the organization is responsible for the integration. The integration module receives as-is information from the input gates and then converts these data into the internal ML scoring extension structures. As such, the mentioned part of the software, in a standardized manner, polls the internal API ML scoring extension. In real implementations, the integration module is custom and cannot be a ready made as a universal "box".

External API: This in a system access module that can be used if the organization can or wants to integrate with external software. The benefit for the organization when using the external API for external gateways is that the data it uses and the services it queries are already in the well-known ML scoring extension format. As such, we ignore the loss of performance in the transformation between different formats and simplify the physical architecture of the solution. The external API issues access using modern technologies using REST API (NodeJS) and AMQP (RabbitMQ). The external API, in calling the Internet API, creates an abstraction layer that clearly separates the core from the implemented systems. This results in the possibility of controlling permissions (the list of ML scoring extension services available for individual internal systems of the organization may be different) and additional verification of the entered data.

Internal API: This is the internal system API. Services have a specific scheme using OpenApi. Owing to the standardization, each internal module of the system coherently communicates with the core module. The Internal API orchestrates core services, completes missing data from various services, and creates more complex core requests.

Core: the main module of the system, written in the CPP language. The core maintains journal operations and executes and controls asynchronous recursive operations. A load balancer for services is built in due to the possibility of each module working in several instances. The core functionality also includes acting as a router for services. The request received from the internal API redirects to the appropriate module.

Reporting module: the module used to generate daily and periodic reports. It allows the issuing of a service that generates any report upon request. It has a dedicated, separate database so that the creation of reports does not interfere with the operation of the entire system.

Scoring module: provides scoring services based on ML solutions. Upon input, the operation is evaluated; upon output, it receives one of three possible responses (OK, NOT-OK, or MANUAL). The scoring module uses the internal API to query the ML execution module, which sequentially starts the processing of decisions into deep and shallow autoencoders.

ML tuning module: the service provider for tuning a new solution, which processes the operations file to tune the autoencoders. The module is based on Keras and Tensorflow.

ML testing module: the module that provides services for versioning autoencoders, verifies processed new adjustments, processes full historical tests, and evaluates performance against historical tests.

ML execution module: the module that provides services for the processing of financial and non-financial operations on autoencoders. In response to the request, the operation similarity float value (predict) is returned. We developed the module using Keras.

Operator interface: the module that provides a web interface for a CSIRT operator. It contains data on processing queues for the entire module and its history, the history of module decisions, the currently analyzed operation, and the "MANUAL" decision list.

Administration interface: the module that provides a web interface for the administrator. It allows the granting of privileges to CSIRT operators, viewing journal logs and operations, and viewing operation statistics. Moreover, it provides information on system performance.

Log viewer module: the module that provides a web interface for viewing logs, based on Kibane, Elasticsearch, and APM. It offers multidimensional viewing of system logs, creating data views, and collecting metrics.

Notification module: the module providing notifications within the ML scoring extension system, e.g., whether the ordered report is ready for viewing/downloading.

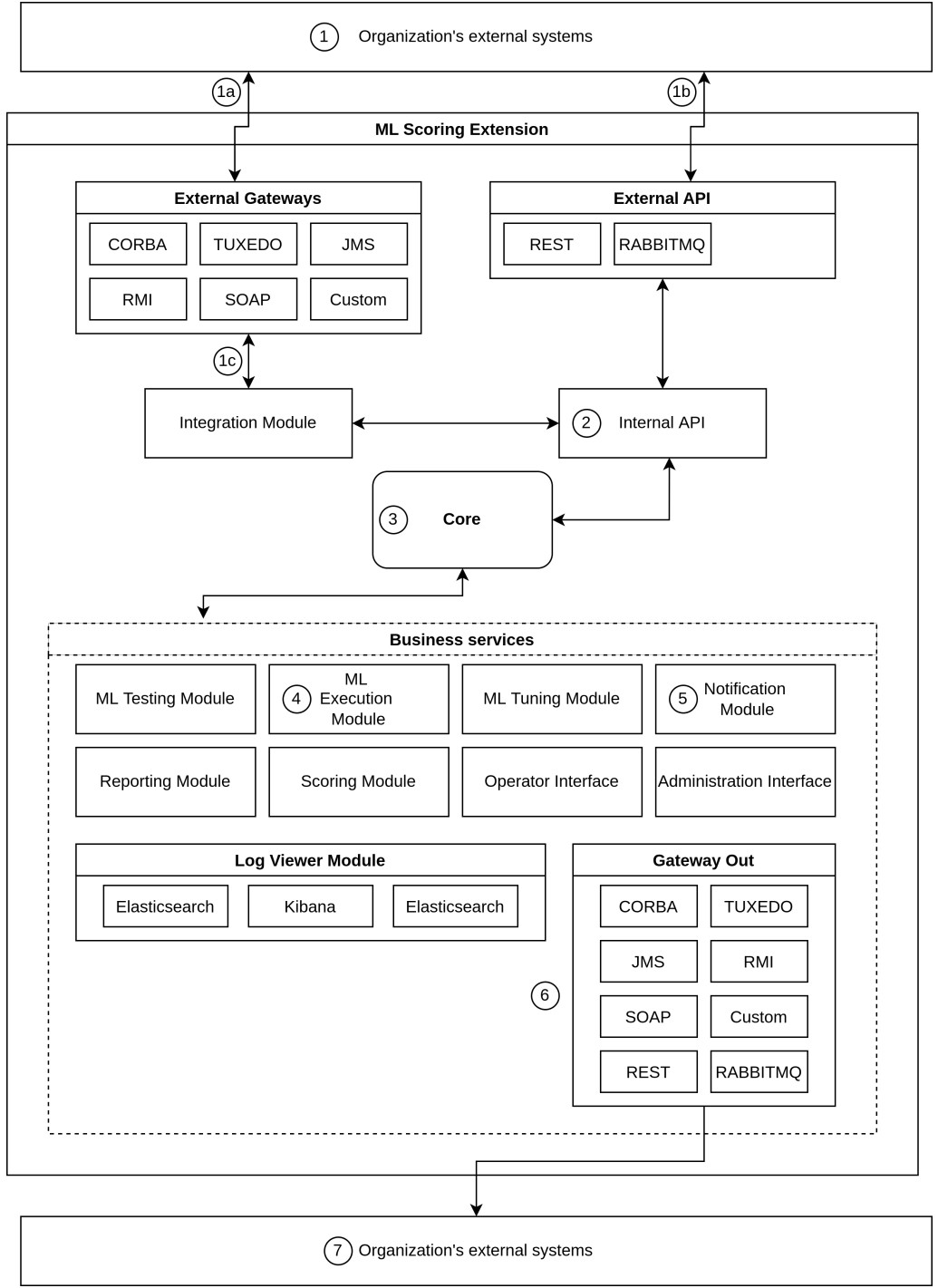

**Figure 2.** ML scoring extension module architecture.

Gateway out: the output gates necessary for communication between ML scoring extension and external systems and for integration with services within the organization (e.g., data warehousing and sending requests to services).

*4.2. Data Flow for Decision-Making Operation in Proposed ML Scoring System*

The data flow for the decision-making operation in the proposed ML scoring extension system is presented in Figure 2. In this figure, one can find the step numbers, which refer to the order of data processing:

Step 1: The organization's core systems trigger a request for a decision in the context of an ongoing financial or non-financial operation:

- 1a: In the event that the organization's systems are not adapted to use the issued API by the ML scoring extension system, technical integration occurs through the interfaces that these systems understand. The request is sent to the external gateway. At this point, there is a contact between the organization's systems and the gates, which translate one transport into the transport understood by the ML scoring extension system.
- 1b: In the event that integration with the organization's systems occurs through development and integration from their side, these systems can directly use the external API. Requests to our system are natively understood by the system, and no extra layer of transformation is needed.
- 1c: Then, the request is sent to the integration module, where the technical and business transformation of the request coming from the organization's systems into an abstraction understood by our system occurs. The business data create a valid request to the internal API native to our system.

Step 2: The internal API verifies the parameters of the operation and authorizations to invoke services. When a decision is made, this service takes the operation upon entry and issues the decision upon exit. It is a high-level service that is completely transparent for the client (they do not know what services are called or in what order).

Step 3: The request is sent to the core from the internal API. The core has data on active instances of the ML execution module and knows what addresses they have and what the load is. The core selects the best instance according to the load-balancing algorithm (choosing the least-loaded, where the request has the fastest chance to process).

Step 4: A financial or non-financial operation is processed. The input vector for Keras/ Tensorflow is created here, which is called in the "predict" method in Tensorflow. On the basis of the threshold from the previously trained autoencoder (AE), one of three types of decisions is determined: "OK" (white), "MANUAL" (gray), or "NOT OK" (black).

Step 5: When notified that the decision is "MANUAL" (gray) or "NOT OK" (black), the ML execution module sends a message via the core to the notification module that an email notification should be sent to the CSIRT. The notification module uses ready-made email templates. If the organization's systems are responsible for sending the message, it forwards the content of such an email to the gateway out.

Step 6: In the gateway out, various support systems of the organization are integrated, in this case with the email-sending service. The previously prepared email is pushed to the service within the organization for further processing and sending the email.

Step 7: The organization's external system performs the received requests in accordance with the agreed functionalities.

## 5. ML Execution Module

In our approach, we created the ML execution module based on data stored during users' logging into the banking system. Based on these data, we designed the ML models described in the following sections.

### 5.1. Data Acquisition and Preparation

We based our ML experiments on unlabeled real-world data of actual login attempts (Table 1) that were collected over two months from the server of one of the largest banks in Poland.

**Table 1.** Characteristics of the training dataset.

| Characteristics of the training dataset | |
|---|---|
| Data collection period | 26 January–31 March 2018 (65 days) |
| Number of records (raw data) | over 5.7 million |
| Number of discarded records (hart-beats) | approximately 3.8 million |
| Number of training records (effective) | 1,918,349 |
| Features extracted (raw) | • Server-side event timestamp<br>• Session ID<br>• Client's IP address<br>• Client's operating system type<br>• Client's browser type and version |
| Features engineered (effective) | • Server-side event timestamp<br>• Client's autonomous system number (ASN)<br>• Client's operating system type<br>• Client's browser type<br>• Client's browser version<br>• Whether client IP is trusted (bank LAN)<br>• Part of the day (working hours, afternoon/evening, or night/early morning)<br>• Working/nonworking days (weekends or holidays) |
| Total number of features (including one-hot encoding) | 36 |
| Training/test split of dataset (%) | 80:20 |

The raw data consisted of over 5.7 million records, from which we filtered out approximately 3.8 million records from technical (heart-beat) logins. We then flattened the remaining records (effectively over 1.9 million) (by parsing the user–agent string) to extract the following features:

- Date and time of the server-side event;
- Session ID;
- Client's IP address;
- Operating system type;
- Browser type and version.

We did not directly obtain the client's IP address from the collected data (because it is often randomly assigned by an ISP), but we used it to find the autonomous system number (ASN) and the client's physical location (geographic latitude/longitude) based on high-accuracy commercial databases. We also discarded the session ID because we did not separate individual logins, but rather, concentrated on the characteristics of the client environment. Additionally, we introduced three virtual features into our dataset (based on the original attributes) to distinguish logins originating from the trusted bank network, the part of the day (working hours, afternoon/evening, and night/early morning), and working days from weekends or holidays. We one-hot-encoded all the categorical variables,

for example the operating system type and the browser type and version, which produced 36 distinct features. We normalized the remaining continuous variables to the [0, 1] range. We randomly split the data into a training set consisting of 80% records; we treated the remaining 20% as a test set.

### 5.2. ML Model Descriptions

We concentrated our subsequent studies on two autoencoder models:

(A)  Classical (AE), which is shallow, consisting only of the input layer *I* with 36 inputs, encoding data vector features, the code (representation) layer *C* with 3 neurons, and the symmetrical output layer *O*, also consisting of 36 neurons (Figure 3);

(B)  Deep AE with additional (also symmetrical) hidden layers $H^1$ and $H^2$, composed of 10 neurons each in the encoder/decoder section (Figure 4).

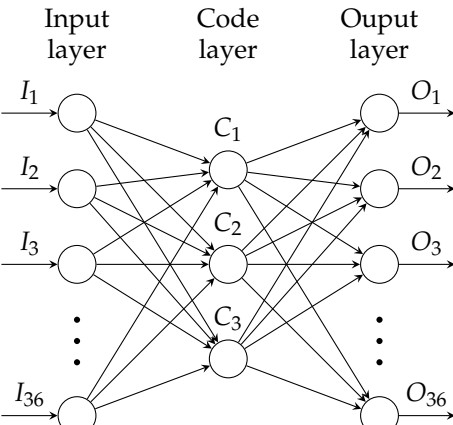

**Figure 3.** Shallow autoencoder model (A).

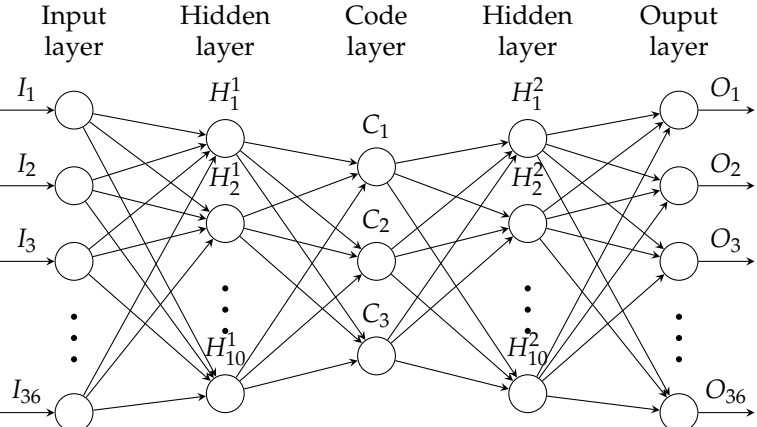

**Figure 4.** Deep autoencoder model (B).

We built the ML models with Keras Version 2.2.4 based on the TensorFlow framework (Version 1.13.1). In both models, we used sigmoid as the activation function in the last (output) layer. For the input and intermediate layers, we tested both ReLU and Swish [31] (with $\beta = 1$) as activation functions for all layers, but the last one in both models. For both models, we finally used the Swish function, because of the faster convergence of the results.

### 5.3. Training Procedure

We trained our shallow autoencoder model (A) for 40–50 epochs to reach convergence, obtaining an MSE of 0.01558 on the test set. The deep model (B) needed substantially more training epochs (200–300) than the shallow model. In both models, we applied no regularization in the code layer of our autoencoders, and we used the adaptive moment

estimation algorithm (Adam) [32] as the optimizer, along with a batch size for training of 1024 records. We chose the mean-squared error (MSE) as the loss function for both models. We also tried other possibilities, such as categorical cross-entropy (log loss) or the $R^2$ score, but we encountered problems with the convergence of our models. The discussion of this finding is beyond the scope of this study, which we will examine separately.

### 5.4. Results

After training, with our shallow autoencoder model (A), we obtained the total value of the MSE (the measure of reconstruction error) equal to 0.01558 for the test set. The distribution of the reconstruction errors both for the training set and the test set is shown in Figures 5 and 6.

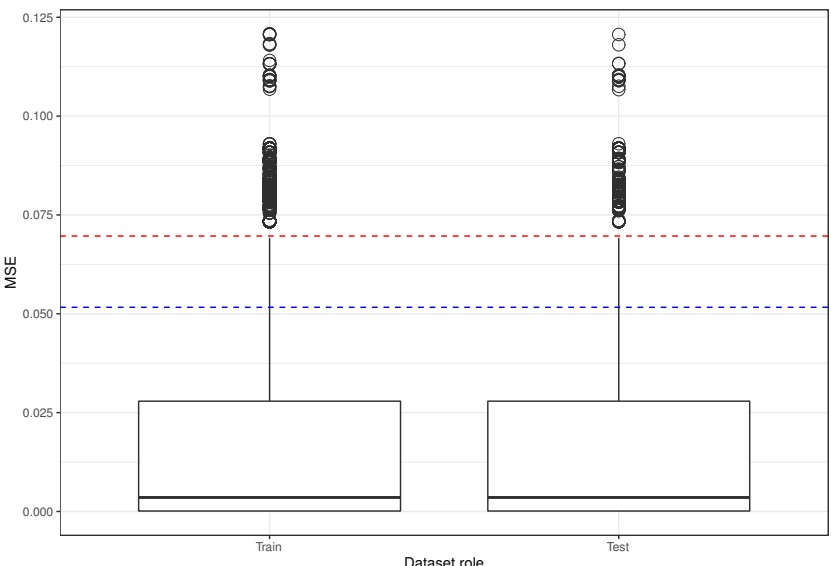

**Figure 5.** Mean-squared reconstruction error (MSE) for Model A (shallow). Blue and red dotted lines mark the $\mu + 2\sigma$ and $\mu + 3\sigma$ limits, respectively.

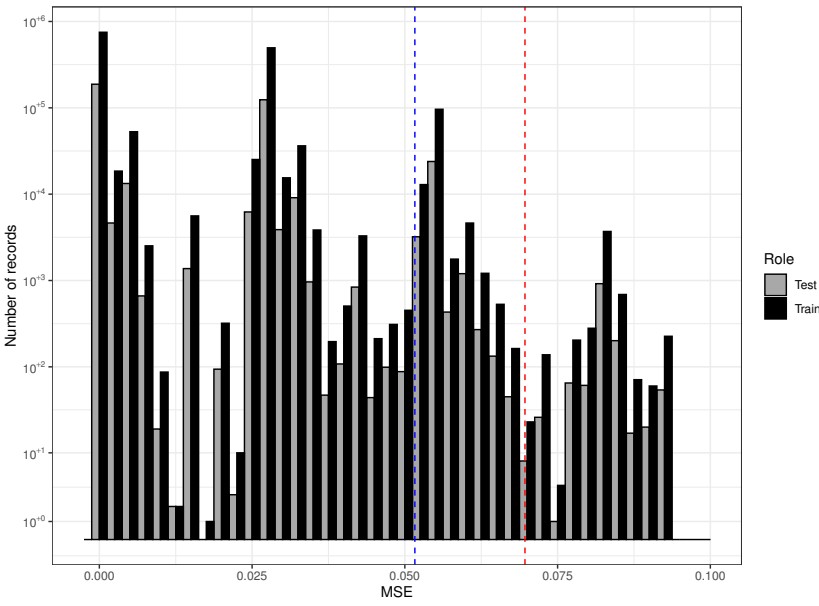

**Figure 6.** Histogram of the MSE distribution for Model A (shallow). Blue and red dotted lines mark the $\mu + 2\sigma$ and $\mu + 3\sigma$ limits, respectively. Note the logarithmic scale on the y-axis.

The mean value $\mu$ of the combined data is equal to 0.0156, and the standard deviation $\sigma$ is 0.0180. One can notice quite a high number of outliers in the boxplot (which could be, potentially, anomalous events), corresponding to high-value MSE bins in the histogram.

For the deep autoencoder model (B), we obtained a total reconstruction error (MSE) of 0.0127, which is smaller than that of the shallow model and indicates its higher restoration capacity and higher representation learning power due to its more complicated structure. The distribution of events and their MSE values (with a mean value $\mu$ of the combined data of 0.0127 and a standard deviation $\sigma$ of 0.0164) are shown in Figures 7 and 8.

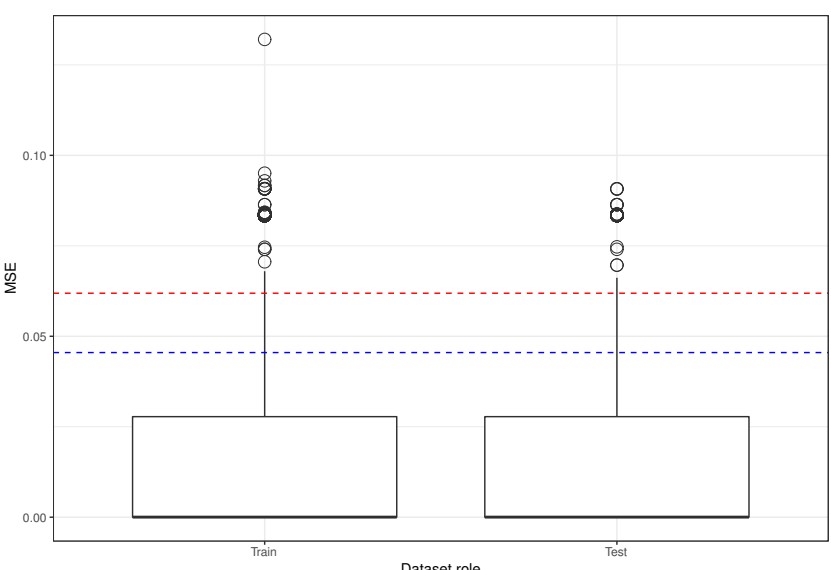

**Figure 7.** Mean-squared reconstruction error (MSE) for Model B (deep). Blue and red dotted lines mark the $\mu + 2\sigma$ and $\mu + 3\sigma$ limits, respectively.

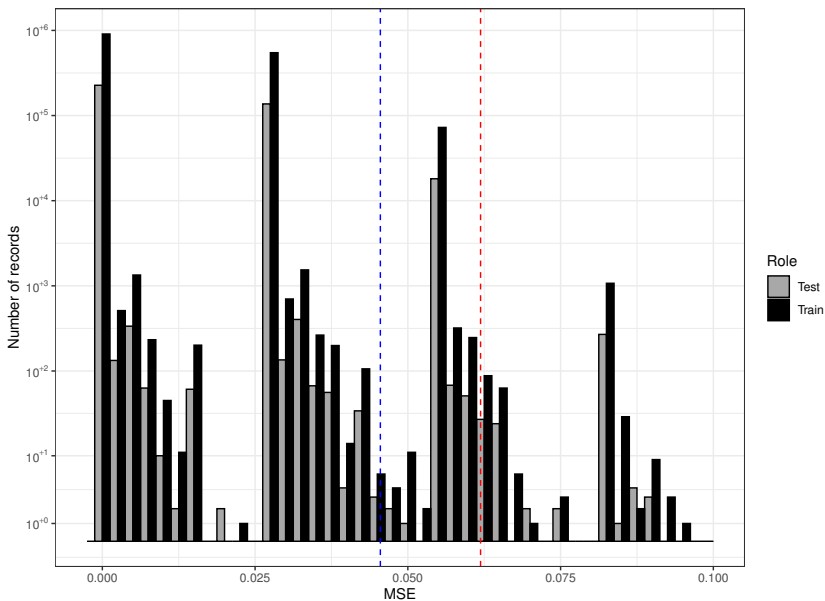

**Figure 8.** Histogram of the MSE distribution for Model B (deep). Blue and red dotted lines mark the $\mu + 2\sigma$ and $\mu + 3\sigma$ limits, respectively. Note the logarithmic scale on the y-axis.

This time, the high-value MSE bins are invisible in the histogram, which was due to the substantially fewer outliers in the boxplot than in the shallow model.

We interpreted these results, after analyzing the boxplots and histograms presented in Figures 5–8, as follows (see Table 2): The first cluster of more than $10^6$ cases with an

MSE close to zero most probably represents regular events, resulting from legitimate logins executed by the bank customers. The second cluster, with MSE values of approximately 0.030 (accounting for approximately $6 \times 10^5$ elements), could be an effect of a small (perhaps one-variable) change in a customer's record (e.g., upgrading to a newer version of his/her browser). They lie within the $\mu + 2\sigma$ limit (which is why they are usually denoted as white cases) and contained 1,765,415 (92.03%) for Model A and 1,825,370 (95.15%) for Model B of all the records.

These results, after analyzing the boxplots and histograms presented in Figures 5–8, can be interpreted as follows (see Table 2): The first cluster of more than $10^6$ cases with the MSE close to zero represents most probably regular events, resulting from legitimate logins, executed by the bank customers. The second cluster, with an MSE of approximately 0.030 (with approximately $6 \times 10^5$ elements), could be an effect of a small (perhaps one-variable) change in a customer's record (e.g., upgrading to a newer version of his/her browser). They lie within the $\mu + 2\sigma$ limit (which is why they are usually denoted as white cases), which contained 1,765,415 (92.03%) for Model A or 1,825,370 (95.15%) for Model B. The third cluster, containing slightly more than $10^5$ events with an MSE of approximately 0.06, could be attributed to slightly larger variations in the login data, such as access attempts from some irregular geographic location (such as during holidays or a business trip) or an unusual time of day. These events (counting 146,146, i.e., 7.62% for Model A, and 91,374, i.e., 4.76% for Model B) were clearly discriminated by our model (with the deep version of Model B performing better). However, due to their abundance and presence within the $\mu + 3\sigma$ limit, they could not actually be regarded as really anomalous cases; notably, they may need further attention and can be regarded as gray cases. Only those lying above $\mu + 3\sigma$, i.e., 0.0697, for the shallow model (6788 records, accounting for only 0.35% in total) and 0.0619 for the deep model (1605 records, again accounting for only 0.08% of the total), should be treated and further analyzed as suspicious (black) events and/or verified by another, independent authentication channel. Therefore, in the latter case (Deep Model A), the burden of the manual discrimination of potentially fraudulent actions can be reduced to less than five instances per day.

**Table 2.** Number and percentage of events that fall into the white, gray, and black categories.

| Model | White | | Gray | | Black | |
|---|---|---|---|---|---|---|
| | # | % | # | % | # | % |
| Shallow (A) | 1,765,415 | 92.03 | 146,146 | 7.62 | 6788 | 0.35 |
| Deep (B) | 1,825,370 | 95.15 | 91,374 | 4.76 | 1605 | 0.08 |

## 6. Conclusions and Future Work

In this paper, we presented a new architecture for an antifraud system with an ML scoring extension module. We described the modules that participate in the architecture and the data flow. We designed an ML-based scoring extension module, which is responsible for decision-making, classifying operations as white, gray, or black. We developed an unsupervised method to distinguish between rogue and legitimate bank account login attempts, using two autoencoder models (shallow and deep), which we trained to detect bank fraud attacks based on real data. The obtained results indicated that the proposed autoencoders can be used as an ML-based scoring system for an antifraud system in the banking environment. Our system is based on data that can be gathered at an early stage, before the transaction is defined by the user.

Basic antifraud systems detect the most-suspicious and largest transactions, but the scale of small crimes generates the largest losses. The proposed antifraud system can financially benefit institutions because it can detect potentially harmful operations, which are forwarded to the CSIRT for verification. These operations are identified before the actual financial operation begins, already at the login stage. This produces the effect of

eliminating the risk of low-value operations, which were previously included in the costs of running the organization.

In future studies, we will expand the system with behavioral models that can further address the issue from the customer side. This involves profiling and identifying the customer by how he/she navigates the system for finding suspicious activities.

**Author Contributions:** Conceptualization, M.S., A.B. and B.K.; methodology, M.S., A.B. and B.K; software, M.S.; validation, M.S. and A.B.; formal analysis, M.S. and A.B.; resources, M.S.; data curation, M.S.; writing—original draft preparation, M.S., A.B. and B.K.; writing—review and editing, M.S., A.B., B.K. and M.W.; visualization, M.S. and A.B.; supervision, M.W. All authors have read and agreed to the published version of the manuscript.

**Funding:** This study received no external funding.

**Institutional Review Board Statement:** Not applicable.

**Informed Consent Statement:** Not applicable.

**Data Availability Statement:** 3rd Party Data.

**Conflicts of Interest:** The authors declare no conflict of interest.

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
