# Peer review of "Machine-Learning-Based Scoring System for Antifraud CISIRTs in Banking Environment"

_electronics, doi:10.3390/electronics12010251_

Round 1
Reviewer 1 Report
I suggest to format the dataset description in a table consisting of the total number of samples along with the features and the other specifications.
Other metrics other than MSE should be considered for evaluation.
English must be improved especially the verbs tense. Examples includes (We proposed, We presents, In the article we would like to propose, etc.)
Author Response
Dear Reviewer,We are pleased to resubmit for publication the revised version of the
article entitled Machine-learning-based scoring system for antifraud CISIRTs
in banking environment. We appreciate your constructive criticism.
We have addressed each of your concerns in attached pdf file.
Best Regards,
Bogdan Ksiezopolski

Reviewer 2 Report
In this manuscript, a new architecture for an anti-fraud system with an ML scoring extension module is presented. The scheme proposed in the paper seems sound and the organization is easy for me to follow it. However, I have some suggestions and questions on this work.
1. The paper does not seem very detailed in describing the innovation points.
2. Although ML is well known, it is still recommended to give the full name in this paper.
3. Should privacy leakage be considered in the anti-fraud data collection process?
4. It is recommended that Figure 6 select an appropriate scale and redraw in detail.
Author Response
Dear Reviewer,
We are pleased to resubmit for publication the revised version of the article entitled Machine-learning-based scoring system for antifraud CISIRTs in banking environment. We appreciate your constructive criticism. We have addressed each of your concerns in attached pdf file.
Best Regards,
Bogdan Ksiezopolski
